# Topical Administration of *Lactiplantibacillus plantarum* (SkinDuo^TM^) Serum Improves Anti-Acne Properties

**DOI:** 10.3390/microorganisms11020417

**Published:** 2023-02-07

**Authors:** Christine Podrini, Laetitia Schramm, Giulia Marianantoni, Jagienka Apolinarska, Colin McGuckin, Nico Forraz, Clément Milet, Anne-Laure Desroches, Pauline Payen, Maria D’Aguanno, Manuele Biazzo

**Affiliations:** 1The BioArte Ltd., Malta Life Science Park (LS2.1.10, LS2.1.12, LS2.1.15), Triq San Giljan, SGN 3000 San Gwann, Malta; 2Cell Therapy Research Institute, CTIBIOTECH, Bat A16, 5 Avenue Lionel Terray, Meyzieu, 69330 Lyon, France

**Keywords:** acne, probiotic, inflammation

## Abstract

The tailoring of the skin microbiome is challenging and is a research hotspot in the pathogenesis of immune-mediated inflammatory skin diseases such as acne. Commonly encountered preservatives used as functional ingredients have an impact on the skin microbiota and are known to inhibit the survival of skin commensal bacteria. The selected species is *Lactiplantibacillus plantarum*, formulated with natural enhancers for topical use (SkinDuo^TM^). Ex vivo human skin models were used as a test system to assess the strain viability which was then validated on healthy volunteers. SkinDuo^TM^ showed increased viability over time for in vitro skin models and a stable viability of over 50% on healthy skin. The strain was tested on human primary sebocytes obtained from sebaceous gland rich areas of facial skin and inoculated with the most abundant bacteria from the skin microbiota. Results on human ex vivo sebaceous gland models with the virulent phylotype of *Cutibacterium acnes* and *Staphylococcus epidermidis* present a significant reduction in viability, lipid production, and anti-inflammatory markers. We have developed an innovative anti-acne serum with *L. plantarum* that mimics the over-production of lipids, anti-inflammatory properties, and improves acne-disease skin models. Based on these results, we suggest that SkinDuo^TM^ may be introduced as an acne-mitigating agent.

## 1. Introduction

Acne remains the most common inflammatory dermatosis treated worldwide. According to the Global Burden of Disease, the prevalence rate of acne is 9.4% [1], making it the eight-most prevalent disease in the world [2]. Adolescent and young adults (12 to 24 years of age) show among the highest prevalence rate of acne (estimated to be up to 85%), but the disease can persist beyond young adulthood despite treatment [3]. The clinical presentation of acne varies from primarily comedonal to mixed comedonal and inflammatory acne. Acne is categorized according to severity: comedonal acne, which consists of small white papules (closed comedones), or grey and white papules (open comedones), which are due to complete or partial ductal occlusion, respectively, and sebum accretion; mild-to-moderate papulopustular acne, which is characterized by inflammatory lesions that are mostly superficial; and severe acne, consisting of deep pustules and/or nodules, which may be painful, may extend over large areas, and can lead to tissue destruction [4,5]. Acne is a multifactorial complex disease of the pilosebaceous unit that can be approached with different medication regimes based on type of acne, location, severity, and the feasibility of treatment management [2]. Although acne is not a life-threatening disease, it is associated with low quality of life and may lead to emotional stress and scarring [6].

Acne vulgaris is a chronic and relapsing inflammatory skin disease, mainly caused by *Cutibacterium acnes* (*C. acnes*) overgrowth that might increase lipid production and has long been proposed to be the most prominent microorganism leading to inflamed acne lesions. There is increasing evidence to suggest that not only *C. acnes* but also *Staphylococcus epidermidis* (*S. epidermidis*), and the whole skin microbiome play key roles in the development of acne. Both strains of bacteria are present in acne lesions [7]. However, between 55% to 68% of all isolates showed a loss of *C. acnes* phylotype diversity, with a predominance of phylotype IA1 [8] and a decrease of phylotypes IB and II strains [9].

Acne, besides being an inflammatory skin condition, it involves an interplay of several factors, including the abnormal keratinisation of the sebaceous canal, bacterial colonisation [10], increased sebum production, genotypic factors, and hormonal disorders. Another factor associated with acne development is the interactions between the microbiome and the host innate immune system. The intestinal microbiota is involved in the formation of acne lesions [11], and it also plays a key role in the development and maintenance of a proper immunity of the host organism. Acne is a disease that can be related to the condition of the digestive tract and its microbiome. The existence of a gut-skin axis is supported by increasing evidence and intense research, but its translational potential is yet not recognized. As studies have linked inflammatory skin diseases to imbalanced skin and/or intestinal microbiota, restoring healthy microbiomes could constitute an interesting therapeutical approach to improve dermatological conditions [12]. Today, there is a growing interest in understanding the role of the skin microbiome in acne development. Investigating the skin microbiome is highly challenging, notably due to technical limitations (including but not limited to low sample biomass, high sample contamination, and sample collection methods). Although studies principally focused on the gut microbiome, bacterial communities from the skin play essential roles in the development of the host immune system and in the protection against invading pathogens and/or other exogenous substances.

The grading of the severity of acne determines the appropriate treatment [13]. Treatment of acne should be started as early as possible to minimise the risk of scarring and adverse psychological effects, which may persist long into adulthood. Topical agents are the mainstay for treatment, many of which are ineffective, targeting keratinocyte turnover, exfoliation or surface sterilisation which is disruptive to the natural skin microbiome. Moderate acne is treated with oral and topical antibiotics, which also reveal a variability of treatment success. More persistent acne, particularly in adolescent boys, may require stronger strategies including topical retinoids [3]. Treatment is often long and tedious and can lead to a reduction in quality of life and social isolation. Additionally, because of the rising problems of antibiotic resistance, alternatives are needed. Acting on the skin microbiome to restore a healthy composition is highly challenging. A promising approach could be the use of live microorganisms or ‘probiotics’ that when administered in adequate amounts will confer a health benefit on the host [14].

Here, were studied the effects of the application of live bacteria on a subtype of the most common inflammatory skin disorders (acne vulgaris) defined as papulopustular acne. We investigated the potential of topically applied live lactobacilli (SkinDuo^TM^) to beneficially modulate cutaneous microbial interactions and inflammatory responses in acne skin models. We successfully developed a probiotic serum containing live lactobacilli (*L. plantarum)* which remain alive when applied to the skin.

## 2. Materials and Methods

### 2.1. Skin Samples

The research described in this work was performed in adherence with the Declaration of Helsinki. Samples, sebocytes and skin used in this study was provided from human skin samples following ethical consent after elective surgery. Human skin tissue was purchased from Genoskin (Toulouse, France) under French law (L. 1245 CSP), as “a product and element of body taken during a surgical procedure and used for scientific research,” with the patients’ consent, and no further ethical approval was required. Skin samples, which were surgical waste, were transported under optimal conditions to ensure and maintain tissue viability and with certification (AC-2018-3243 and DC-2018-3242) by the French Ministry in charge of research for the preparation and conservation of elements derived from the human body.

### 2.2. Bacterial Growth

Cultures from swab: the collected swab was suspended in 2 mL sterile water and then diluted 10^4^ and spread on MRS plates (ThermoFisher Scientific, Waltham, MA, USA).

Strains of lactobacilli were grown at 37 °C in de Man, Rogosa and Sharpe (MRS) medium (ThermoFisher Scientific, Waltham, MA, USA) and cultured micro aerobically (5% CO_2_) at 37 °C for 24–48 h. Time-course experiments were performed on the selected *Lactobacillus* strains, and colony forming units (CFU) were counted.

### 2.3. Tissue Culture

Skin models (NativeSkin^®^) with an 8-mm diameter were obtained from Genoskin (Toulouse, France). Upon arrival, the models were directly transferred into six well plates and allowed to recover overnight in 1 mL of antibiotic free skin medium at 37 °C and 5% CO_2_ as recommended by the manufacturer. Following recovery, the skin models were subjected to the topical application of reconstituted serum in a humidified atmosphere for a total of 48 h. Bacterial growth from skin models was quantified by CFU/mL for each time points tested: 0 h, 8 h, 24 h and 48 h.

### 2.4. Human Skin Cell Culture Co-Culture with Bacterial Strains

Human primary sebocytes (CTIBIOTECH, Lyon, France) were seeded, 3 days before inoculation, in 24-well plates coated in fibronectin (15,000 cells/well) in sebocyte growth and differentiation medium (Seb4Gln, CTIBIOTECH, Lyon, France) without antibiotic, so as not to impede bacterial growth.

### 2.5. Bacterial Strains Selection and Amplification

The commercial strain of *Lactiplantibacillus plantarum* (International Collection Deposit Number: LMG P-21021, Commercial Code LP01) used in the serum was selected, amplified, and characterized by Probiotical Research S.R.L., Novara, Italy. The strain *L. plantarum* LP01 was isolated from a healthy human and not exposed to any genetic manipulation.

Two commercial reference strains were selected and amplified (Eurofins BactUp, Saint-Priest, France) DSM 1897 (ATCC6919) a phylotype IA strain of *C. acnes* isolated from acne-prone facial skin and CIP28.61 (ATCC12228), a strain of *S. epidermidis* of unknown origin (reference strain for FDA quality control tests). DSM 1897 (ATCC6919 or NCTC 737 (DSMZ)) strain: *C. acnes* type IA strain isolated from acne-prone facial skin, were grown at 37 °C in COS agar (Biomerieux, Lyon, France) anaerobically for 48–72 h.

Suspensions of OD_600_ 0.5 to 1 were made and counted on solid culture medium. Correlations between OD_600_ and CFU were used to determine the parameters for preparing the inoculum.

Tryptone soy agar was used as a classical isolation medium (Sigma-Aldrich, St Quentin Fallavier, France).

### 2.6. Lipid Production Analysis

After 7 days of culture, sebocytes were fixed in formaldehyde 4% *w*/*w* (Sigma-Aldrich) and stained with 50 µL/mL Hoechst and 7 µL/mL Nile Red (Sigma-Aldrich) solutions. Fluorescence intensity (FI) was measured with a fluorometer (TECAN, Lyon, France) in each well with the following excitation and emission wavelengths: Nuclei (Hoechst): excitation 356 nm, emission 465 nm; Neutral lipids (Nile Red): excitation 475 nm, emission 530 nm; Total lipids (Nile Red): excitation 520 nm, emission 625 nm. Raw values were collected with SparkControl™ software and exported in Excel to be analysed. Each condition was carried out in triplicate and at least three readings were achieved in different parts of each well. Error bars represent ±SD. Microscopic observations were carried out with a fluorescence microscope (Eclipse Ti, Nikon, Amsterdam, The Netherlands) and treated with NIS software (NIS-Elements BR 4.13.01 64-bit, Nikon, Tokyo, Japan). For each condition, photos were taken in brightfield and with different excitation and emission filters to reveal nuclei in blue (“DAPI”), neutral lipids in green (“FITC”), and total lipids in red (“TRITC”). A merge of the three pictures taken with the filter was conducted, revealing lipid droplets in yellow.

### 2.7. Metabolic Activity

The metabolic activity assay was performed using an Alamar Blue test (Thermofisher, Lyon, France), which investigates the overall health and metabolic activity of cells and tissues in culture. The Alamar Blue reagent was added to the active cell cultures during timepoint analysis. The fluorescent signal detected is proportional to the active/healthy cells. Before (Day 0) and after (Day 4) serum treatment, culture supernatant was removed and replaced by new medium containing 10% Alamar Blue. Then, after 1 h of incubation, the culture supernatant containing Alamar Blue was removed and fluorescence (Excitation 570 nm—Emission 585 nm) was assessed with a TECAN SPARK^®^ fluorometer.

### 2.8. Interleukin Production

ELISA assays were performed for IL-1α, IL-6, IL-8 measurement. This type of assay was appropriate to understand the underlying immune system involvement. At 48 h after serum application to the co-culture (sebocytes/bacteria), culture supernatants were collected and stored at −80 °C. After thawing, the interleukin assay was performed using commercial tests (IL-1α, Antibodies Online, Aachen, Germany; Il-6 and IL-8, Abcam, Cambridge, UK). To obtain the most accurate quantities, each ELISA test was performed twice: a first time to find the appropriate sample dilution to fit in the standard curve on control conditions, and a second time on all the conditions with the previously defined dilution. The manufacturer’s instructions were applied for calculations of the output of cytokines.

### 2.9. qPCR and qPCR-PMA Assay

DNA extraction was performed with a MagMAX^TM^ Microbiome Ultra Nucleic acid isolation kit (Applied Biosystems, Waltham, MA, USA): briefly, 800 µL of lysis buffer was added to 400 µL of sample and lysed for 10 min at 2500 rpm. It was then centrifuged for 2 min at 14,000× *g* and transferred the supernatant (500 µL).

The species of *Lactobacillus* was determined by species-specific PCR identification.

The *Lactobacillus* gene was amplified using primer pairs (F:5′ GAAACCTACACACTCGTCGA 3′; R:5′ CCTGAACTGAGAGAATTTG 3′) with the Luna^®^Universal qPCR Master mix protocol (New England BioLabs, Ipswich, MA, USA) as per the manufacturer’s instructions. A BioRAD CFX96 C1000 touch was used for detection.

A qPCR with viability dye PMAxx (Biotium, Freemont, CA, USA) was used for a subset of experiments, as indicated in the text. Following the manufacturer’s instructions, dead control samples were heat inactivated at 70 °C for 5 min, with (untreated) and without viability dye (treated). The working solution for the dark final was 25 uM. Samples were exposed to light for 15 min, followed by the extraction of genomic DNA. A qPCR was performed with the *L. plantarum* primers. The percentage of viable cells was determined as follows:dCT_sample_ = Ct (_sample, dye treated_) − Ct (_sample, untreated_)

The absolute number of viable cells in samples and standard qPCR was calculated from the standard curve using genomic DNA from *L. plantarum*, and the copy number of the sample was calculated as follows:Ct = slope + y-intercept
Cell number _sample_ = (Ct-yintercept)/slope

For qPCR standards, the absolute number of viable cells in samples and standard qPCR was calculated from the standard curve using genomic DNA from *L. plantarum*.

### 2.10. Sample Collection and DNA Extraction

Skin samples were collected by brushing the forearm and/or the forehead with a sterile swab (Deltalab, S.L. Barcelona, Spain) soaked in sterile water over an area of 2 cm by 2 cm for 30 s. Swabs were transferred to 1.5 mL Eppendorf tubes and stored at 4 °C until extraction (within the same day).

DNA was extracted with a MagMAX^TM^ Microbiome Ultra Nucleic acid isolation kit (Applied Biosystems, Waltham, MA, USA) using a KingFisher Flex Purification System (Thermo Fisher Scientific, Waltham, MA, USA). The subsequent steps of the DNA extraction were executed according to the manufacturer’s instructions. The DNA concentration in the extracts was determined by a Nanodrop^TM^ 8000 spectrophotometer (ThermoScientific, Waltham, MA, USA).

### 2.11. Irritation Patch Skin Test

This study was carried out to assess any potential side effect (skin erythema and oedema reactions) that may occur after applying a SkinDuo^TM^ serum product and was performed independently by Complife S.r.l., Nutratech headquarter, Cosenza, Italy.

### 2.12. Statistics

An unpaired *t*-test was performed with Prism 9.5.0, GraphPad Software. GraphPad style reports: 0.1234 (NS), 0.0332 (*), 0.021 (**), 0.0002 (***), <0.0001 (****).

## 3. Results

### 3.1. Rationale for Lactiplantibacillus plantarum

A thorough screening approach was applied based on the rationale that the strains need to be safe [15], be robust, and have the capacity to exert beneficial functions on the human skin, including skin microbiome modulation [16], immune modulation [17] and epithelial barrier enhancement [18,19]. The species selected, *Lactiplantibacillus plantarum,* (previously known as *Lactobacillus plantarum* [20]) was chosen for robustness and growth capacity. Firstly, the impact of the combined components found in the SkinDuo^TM^ serum that included mannitol, hyaluronic acid and vitamin B1 on *L. plantarum* viability was assessed by comparing it to the viability of the *L. plantarum* solemnly. The SkinDuo^TM^ dry formulation and the dry *L. plantarum* alone were prepared and stored at room temperature. At each defined time point (Appendix A), the SkinDuo^TM^ dry formulation and dry *L. plantarum* were reconstituted in sterile water. Samples were analyzed using the PMA-qPCR method and the results (in percentage) were reported as the proportion of alive *Lactobacillus* over the total amount of *Lactobacillus* (Appendix A). The proportion of alive bacteria was overall stable over time, in fact an increased number of *L. plantarum* was observed after 7 days and 4 weeks for the SkinDuo^TM^ dry formulation when compared to *L. plantarum* only (Appendix A), suggesting an enhanced viability. The enhanced viability might be due to the bacteria utilizing the formula ingredients to support their growth.

The viability of *L. plantarum* in solution (mannitol, hyaluronic acid and vitamin B1) in the SkinDuo^TM^ formulation was evaluated over defined time-points (0 h, 8 h, 24 h, 48 h, 72 h, 96 h, 120 h, 1 week, 2 weeks, 3 weeks) at RT and at 4 °C (Figure 1A). Each serum sample was tested for viable counts at the defined time points by sample plating followed by colony counting on MRS agar. The results were reported in the CFU/mL of the serum plated. For each time point, the proportion of alive *L. plantarum* was expressed as a percentage of alive bacteria compared to the initial number of alive bacteria (at time 0) (Figure 1A).

The enumeration of the viable population was stable up to 1 week after reconstitution of the SkinDuo^TM^ serum (Figure 1A). However, a decrease of 50% was observed after 24 h of reconstitution, which was more pronounced at 4 °C. This decrease could be due to the slow adaptation of the bacteria to the environment, resulting in lower survival. To validate the enumeration results obtained and to confirm the decrease at 24 h, a viability quantitative PCR (qPCR) utilizing propidium monoazide (PMA) was used. The SkinDuo^TM^ dry formulation was solubilized in sterile water, kept at room temperature, and samples were collected at the times 0, 24 h, 48 h, 72 h, and 7 days after serum reconstitution for PMA-qPCR determination. The quantification was based on a standard curve obtained from the ten-fold serial dilution of extracted DNA from an *L. plantarum* pure culture. The advantage of using PMA-qPCR [18] is that it allows for the selective amplification of DNA from living *L. plantarum*. The obtained results were expressed as the percentage of live bacteria, with time 0 (0 h) being the reference and each time point being reported relative to 0 h (Figure 1B). As previously shown using colony counting methods (Figure 1A), a 50% decrease of live *L. plantarum* was observed after 24 h in the reconstituted SkinDuo^TM^ serum (Figure 1B). The percentage of live bacteria at 48 h, 72 h and 7 days remained stable and exceeded 50% viability when compared to time-point 0 (Figure 1B), similarly to Figure 1A.

Furthermore, the SkinDuo^TM^ safety was assessed by performing a skin irritation test on 25 healthy volunteers using a skin patch test and graded according to Berger et al. [21]. No erythema, oedema or irritation was observed in any of the volunteers tested (Appendix A). The mean irritation index was <0.5 in the tested subjects, which is between 0.25 and 1 in normal skin [22].

### 3.2. Impact of Lactobacilli Serum on the Skin

We designed a topical formulation for the application of live bacteria in a dose suitable for skin, SkinDuo^TM^. The selected bacteria and formulation were embedded in a core compartment syringe holder. The optimized conditions resulted in a core of dehydrated bacteria that can be released upon the application of pressure and reconstituted in contact with sterile water. The topical serum SkinDuo^TM^ was applied onto hyposkin^®^ models [23] that contained donated human skin biopsies and tested viability up to 48 h of treatment. The impact of the serum with *L. plantarum* on viability was tested after times 0, 8 h, 24 h and 48 h, and compared to control serum (SkinDuo^TM^ only). To test the viability, the samples were plated on selective MRS agar for lactobacilli. As the hyposkin^®^ models had an 8 mm external diameter, CFU/mL represented a standardised area tested, and no further normalization was required. The time-points tested increased per time-point enumeration, indicating metabolically active lactobacilli. The number of lactobacilli persisted and doubled over time and reached statistical significance at 48 h after SkinDuo^TM^ treatment (Figure 2A and Appendix A).

Furthermore, the topical formulation of live bacteria, SkinDuo^TM^ was applied on skin (10 subjects) in a 2 cm^2^ × 2 cm^2^ area for a total time of 8 h and compared to each baseline at time 0, before serum application. Skin swabs were collected for each time point, and qPCR was determined for the quantification of *L. plantarum* and expressed as log CFU/mL. A resulting 9/10 subjects that applied the serum SkinDuo^TM^ resulted in more than 50% mean viability (log CFU/mL) when compared to initial time 0 (Figure 2B). Serum SkinDuo^TM^ topical application remained alive for 48 h in vitro and up to 8 h in human subjects, as tested and demonstrated.

### 3.3. Lactobacilli Mediates Improvement in Skin Models of Acne Disease

To test the inhibitory activity of lactobacilli against acne-causing bacteria, *C. acnes* was targeted as an in vitro model with inflammatory characteristics of acne vulgaris. Specifically, activity assays against class-A *C. acnes* were conducted, a class that is overrepresented in acne-affected skin [24,25]. *S. epidermidis* was also targeted as an important pathogen causing acne [26]. Sebum produced by sebaceous glands can become dysregulated and may lead to common skin diseases including acne [27]. Human primary sebocytes were isolated and cultured in 2D monolayers and tested for bacterial growth (CFU/mL) in medium specific for *C. acnes* and *S. epidermidis* after 48 h of SkinDuo^TM^ serum application. The positive colonies per strain tested for *C. acnes* and *S. epidermidis* statistically decreased after serum treatment (Figure 3A) in association with human sebocyte contact. As acne appears to participate directly in the augmentation of sebaceous lipogenesis [28], we examined the formation of intracellular lipid in human sebocytes after 48 h of serum treatment. After incubation, the serum treatment on its own decreased the lipid production (43% decrease) when compared to the untreated and the other conditions tested (Figure 3Bi). Furthermore, in the presence of liposaccharide (LPS, which induces the expression of antimicrobial peptides and proinflammatory cytokines), serum treatment remarkably decreased lipid production (57% decrease) when compared to LPS treatment (Figure 3Bi). The immunofluorescent staining with LPS showed a negligible presence of lipids (Figure 3Bii) when compared to untreated and to the bacteria in the presence of LPS (yellow lipid droplets). To examine the effects of antimicrobial activity in terms of cellular activity after serum treatment, the metabolic activity of human sebocytes was measured. Because metabolic activity reflects the inhibition of cellular activity, in the presence of strains tested *(C. acnes* and *S. epidermidis)* and LPS, the treatment with *L. plantarum* serum dramatically increased (42%) the metabolic activity of human sebocytes (Figure 3C). The results are indicative that the serum treatment with *L. plantarum* diminished the production of lipids and enhanced the metabolic activity in a human primary sebocyte harvested from primary human skin. The latter also confirms the absence of toxicity in the model tested after SkinDuo^TM^ serum treatment.

### 3.4. Lactobacilli Targets Inflammation in Human Sebaceous Models of Acne Disease

Inflammation plays a key role in the development of acne vulgaris, and the expression of IL-1 α and IL-8 are significantly upregulated in acne-involved skin. Previous investigations of the role of IL-1 α in acne correlated with a higher level of IL-1 α in the sera of patients even with subclinical stages of acne [29]. The secretion of cytokines IL-1 α, IL-6 and IL-8 was measured after 48 h of infection with *C. acnes* and *S. epidermidis* in human primary sebocytes (same model as before, Figure 3). The levels of secreted IL-1 α, IL-6, IL-8 in the presence of LPS were 578.5 pg/mL, 4827.5 pg/mL, 32,884.9 pg/mL and with serum lactobacilli treatment they decreased to 35.8 pg/mL, 351.8 pg/mL, and 1946 pg/mL respectively (Figure 4). The mean triplicates with error bars representing the standard deviation (Figure 4A–C) are presented. The results were obtained by ELISA, revealing significant levels of decreasing IL-1α, IL-6 and IL-8 after SkinDuo^TM^ serum lactobacilli compared to bacteria only. All together our results suggest that SkinDuo^TM^ serum treatment with live *L. plantarum* targets the production and secretion of pro-inflammatory cytokines that can be used as a therapeutic option for acne vulgaris.

## 4. Discussion

Acne is a skin disease with an inflammatory background. Topical antibiotics represent the most common first-line therapy for acne even though prolonged treatment may not result in an effective cure. However, topical antibiotic agents are most frequently used for mild inflammatory disease, including papulopustular acne. For patients with more severe extensive acne skin disease, oral antibiotics are used together with topical retinoids. The medication is determined according to the severity of the acne [30]. For severe acne, oral isotretinoin (13-cis-retinoic acid) may be used. This medication often results in decreased sebum production and a modulation of inflammatory response. Additionally, oral isotretinoin may lead to prolonged remission, even in the most severe cases. Unfortunately, isotretinoin is associated with a wide range of side effects, including serological lipid profile alterations, mucosal and skin dryness, liver function alteration, and neurological implications [31]. As a result of long-term therapy and severe forms of acne, scars can remain on patient skin and reduce their quality of life. Topical corticosteroids and systemic antibiotics, often used in combination, also constitute alternatives. However, treatments that aim to inhibit skin colonization by *Cutibacterium acnes* and its non-specific eradication often result in the emergence of antibiotic resistant bacteria and the loss of the treatment benefits over time.

Currently, there are an increased number of studies that show the potential use of probiotics and their beneficial effects on the host [32]. Regarding the skin, lactic acid bacteria are the most common type of probiotics used to promote dermatological health [33], and emerging studies have shown that the skin microbiome can play a beneficial role in various skin diseases and improve health conditions. More specifically, *L. plantarum*, known to produce active bacteriocin-like compounds, can inhibit harmful commensal bacteria found on the skin [34]. We propose to use a single strain of *L*. *plantarum* to modulate and improve the host-skin microbiota interactions in subjects with mild-to-moderate papulopustular acne. The novelty of our formulation-therapy relies on the fact that the *Lactobacillus* species will be alive. Viability is necessary for the *Lactobacillus* species to inhibit the colonization of pathogenic microbes and to remodel the health-associated skin commensal bacteria. Our unique and innovative formulation is only composed of natural ingredients that enhance and preserve the *L. plantarum* viability up to 7 days. A daily-mono dose treatment for acne is proposed as a topical overnight application. In agreement with our results, the probiotic viability was maintained for at least 8 h on healthy volunteers. Furthermore, we show that SkinDuo^TM^ is safe for human health, and is non-pathogenic and non-toxic. As acne is a skin disorder of the sebaceous gland that results in an increased sebum production, SkinDuo^TM^ serum was tested on human primary sebocytes that were isolated and cultured in 2D monolayers in medium specific for *C. acnes* and *S. epidermidis*. The results from this study show a decreased lipid production derived from human sebocytes which was further attenuated in the presence of pro-inflammatory mediators. Increased sebum production is associated with acne and utilizes metabolic substrates to promote the growth of *C. acnes* and *S. epidermidis.* After SkinDuo^TM^ serum treatment, the metabolic activity was enhanced, suggesting an improvement of metabolic cellular status which also reflects the inhibition of cellular activity and confirms the absence of toxicity in the acne diseased model.

Importantly, the results from this study demonstrate a mean reduction of 62% of the overrepresented pathogens causing acne: class-A *C. acnes* and *S. epidermidis* after SkinDuo^TM^ treatment when compared to non-treated human primary sebocytes isolated from human primary skin. The nearly complete eradication in the colonization of the harmful bacteria demonstrates the capability of the SkinDuo^TM^ serum as an easily accessible topical treatment for the reduction of the key players of acne vulgaris after only one treatment. The results of SkinDuo^TM^ serum showed a significant decrease (93%) in the production of inflammatory mediators: IL-1α, IL-6 and IL-8, when compared to non-treated human primary sebocytes in the presence of LPS. The high levels of IL-1α have been proven to play a role in acne development in the presence of proinflammatory cytokines [35]. SkinDuo^TM^ treatment prevented the formation of inflammation in an acne-skin diseased model. Among the other inflammatory mediators, IL-8 was shown to modulate the chemokine expression of keratinocytes and to have a decisive role in attracting neutrophils to the pilosebaceous unit [36]. Gene array expression profiling in acne lesions has revealed the marked upregulation of genes involved in inflammation and matrix modeling that included IL-8 [37]. The topical application of SkinDuo^TM^ gives rise to new opportunities for acne treatment that can modulate the inflammatory response, decrease microbial pathogens associated with the disease, and could serve as an alternative to topical antibiotics.

## 5. Conclusions

The results of this study have demonstrated the potential of SkinDuo^TM^ as a new approach to acne management. The generation of an inflammatory process plays a key role in the pathophysiology of acne, and SkinDuo^TM^ is designed to prevent the production of inflammatory mediators. The obtained results show a nearly complete ablation in the formation of IL-1α, IL-8 and IL-6, which subsequently prevents the inflammatory cytokines from mediating innate and adaptive immunity and multiple physiological processes, thus preventing the onset of acne-induced skin inflammation. Additionally, the results showed the inhibition of acne-causing bacteria on human primary sebocytes. The SkinDuo^TM^ serum did not exhibit toxic responses on the skin, and the topical formulation is suitable for the application of live bacteria in a sufficient dose on the skin that is delivered in a viable state. The reduction of inflammatory response, sebum production and the inhibition of acne-causing bacteria is key in acne management.

In conclusion, the SkinDuo^TM^ serum offers a new approach to acne management by targeting the underlying causes of inflammation and inhibiting the growth of acne-causing bacteria. The preservative-free formulation provides a safe and effective option for managing acne while preserving the natural balance of the skin microbiome. Further molecular studies are needed to unravel the underlying immunomodulatory mechanism, but the results of this study will be used as a base to design future clinical studies and to contribute to a new skin therapeutic based on microbiome modulation.

## 6. Patents

M.B. is a co-inventor on a pending patent for the use of the formulation and device for treatments of skin disorders.

## Figures and Tables

**Figure 1 microorganisms-11-00417-f001:**
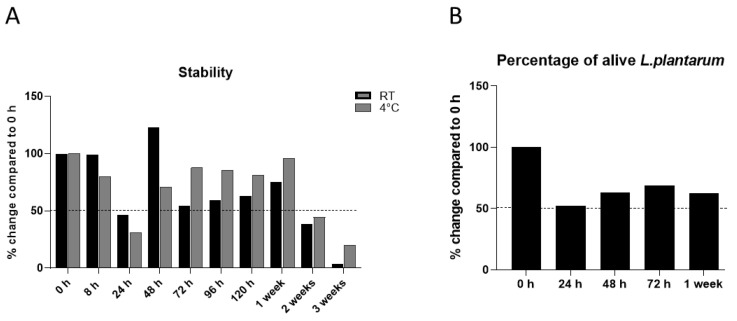
Viability *L. plantarum* expressed as a percentage respective to time 0. (**A**) The serum formulation SkinDuo^TM^ was reconstituted with sterile water and tested for stability as CFU/mL (in percentage) at times 0, 8 h, 48 h, 72 h, 96 h, 120 h, 1 week, 2 week and 3 weeks, and compared to time 0. The viability was maintained over 1 week, except after 24 h and after 2 and 3 weeks. (**B**) The formulation was stored at RT and tested for viability with a qPCR (live dye)—Propidium monoazide (PMA) method, after 24 h, 48 h, 72 h and 1 week. For each condition the ratio of the average CFU/reaction value for PMA-qPCR samples over the average CFU/reaction value for total live bacteria was calculated as a ratio of live bacteria for each condition tested. The percentage change of viability was compared to time 0. Dashed lines represent 50% from time 0.

**Figure 2 microorganisms-11-00417-f002:**
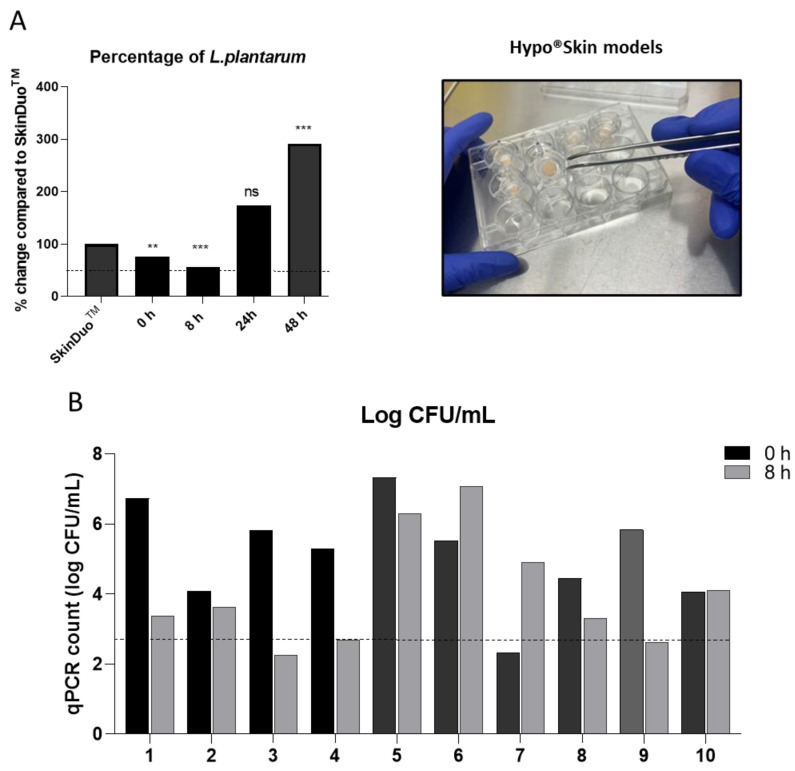
Viability of SkinDuo^TM^ on the skin. (**A**) The serum SkinDuo^TM^ was topically applied to skin biopsies (right picture) and tested for viability at times 0, 8 h, 24 h and 48 h, and compared as a percentage change to serum SkinDuo^TM^ (left graph). CFU/mL was counted in technical triplicates. The mean for each time-point is presented. From time 8 h to time 48 h, there is a doubling increase of *L. plantarum* survival which increases steadily over time. This long-term viability increases the action of *L. plantarum*, without any damage on the skin. The experiment was repeated twice. For time 24 h, the repeated experiment resulted in an increased trend, ns *p* > 0.05 (Appendix A). Statistical comparison (CFU/mL): serum Vs. 0 h, 8 h, 24 h, 48 h; NS. *p* > 0.05; Unpaired *t*-test: 0.021 (**), 0.0002 (***). (**B**) Topical SkinDuo^TM^ was applied on the skin of 10 healthy volunteers (fore-head and fore-arm) for 8 h. Skin swab samples were collected at time 0 (before application) and at 8 h after serum application. Swabs were processed after collection and viable counts were detected by qPCR. The observed value was calculated from a standard curve equation in qPCR and presented as log CFU/mL for the 10 subjects. The dashed line presents the 50% mean log CFU/mL derived from time 0 in all subjects. In 9/10 subjects, more than 50% mean was observed at time 8 h when compared to time 0.

**Figure 3 microorganisms-11-00417-f003:**
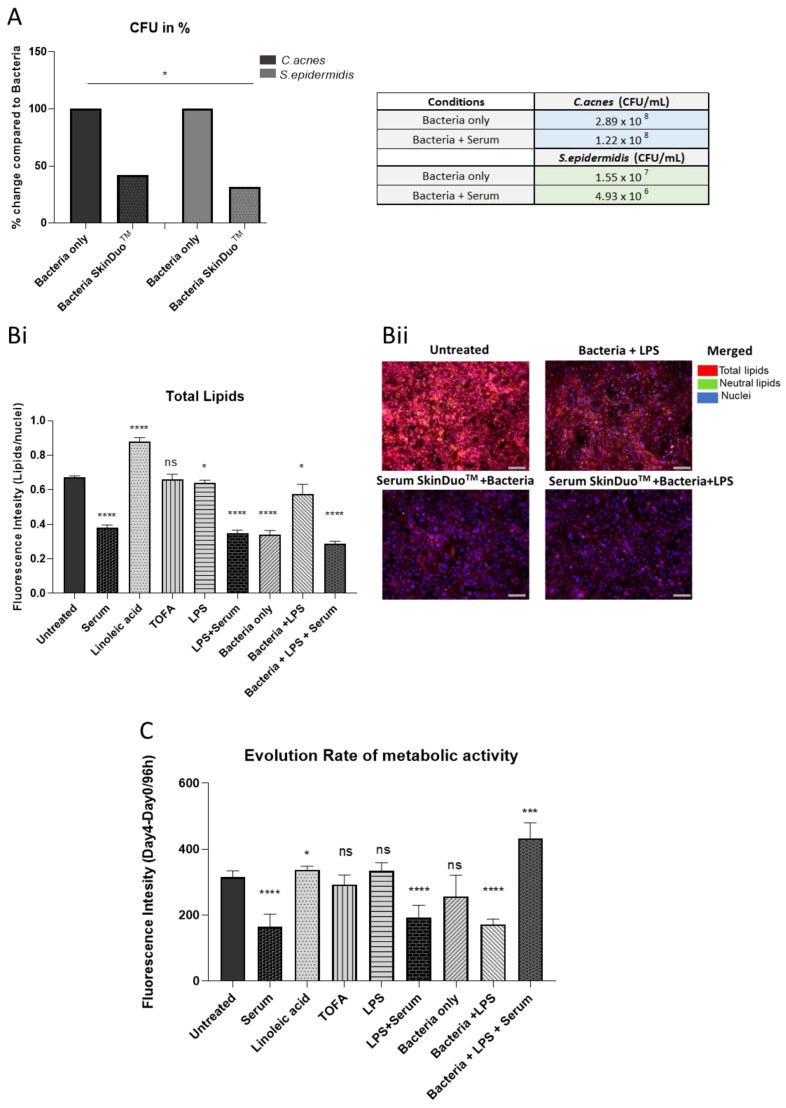
Ex vivo acne sebaceous gland model with virulent phylotype of *C. acnes* and *S. epidermidis* after 48 h of serum SkinDuo^TM^ treatment. (**A**) Bacterial association with human sebocytes shows result per strain (black bar: *C. acnes*, grey bar: *S. epidermidis*). Bacterial enumeration as CFU/mL and expressed as percentage compared to bacteria only showed a decrease in CFU/mL after serum SkinDuo^TM^ treatment, * *p* < 0.05. (**Bi**) Conditions tested in 2D human sebocytes for the following: untreated (basal), serum (SkinDuo^TM^), linoleic acid (positive control for lipid production), TOFA (5-(tetradecyloxy)-2-furoic acid), negative control that reduces fatty acids synthesis and induces inhibition, LPS (lipopolysaccharide) induces an inflammatory challenge; serum in the presence of bacteria (*C. acnes*, *S. epidermidis*) and challenged with LPS. The in vitro cellular model challenged with LPS in the presence of bacteria and treated with serum SkinDuo^TM^ significantly reduced total lipid production when compared to bacteria and challenged with LPS. At baseline, the in vitro cellular model treated with serum decreased lipid production. Fluorescence intensity measurement as a ratio of total lipids/nuclei. (**Bii**) Lipid production analysis was measured with a fluorometer (TECAN) in each well when stained with Hoechst for nuclei ‘DAPI’, Nile Red for neutral lipids ‘FITC’ and total lipids ‘TRITC’, presented as merged picture, lipid droplets are in yellow, and the scale bar represents 20 µm. (**C**) The evolution rate of metabolic activity over time fluorescence intensity measured with Alamar Blue (Day 4–Day 0/96 h). The reduced fluorescence indicates the inhibition of cellular activity in the presence of bacteria and LPS, the metabolic activity was increased after serum treatment. Evolution rate of metabolic activity, unpaired *t*-test untreated (sebocytes) Vs: serum, linoleic acid, TOFA, LPS, bacteria only, bacteria + LPS, bacteria + LPS + serum. Each condition was tested in six technical replicates. Error bars represent ±SD. Unpaired *t*-test: 0.1234 (NS), 0.0332 (*), 0.0002 (***), <0.0001 (****).

**Figure 4 microorganisms-11-00417-f004:**
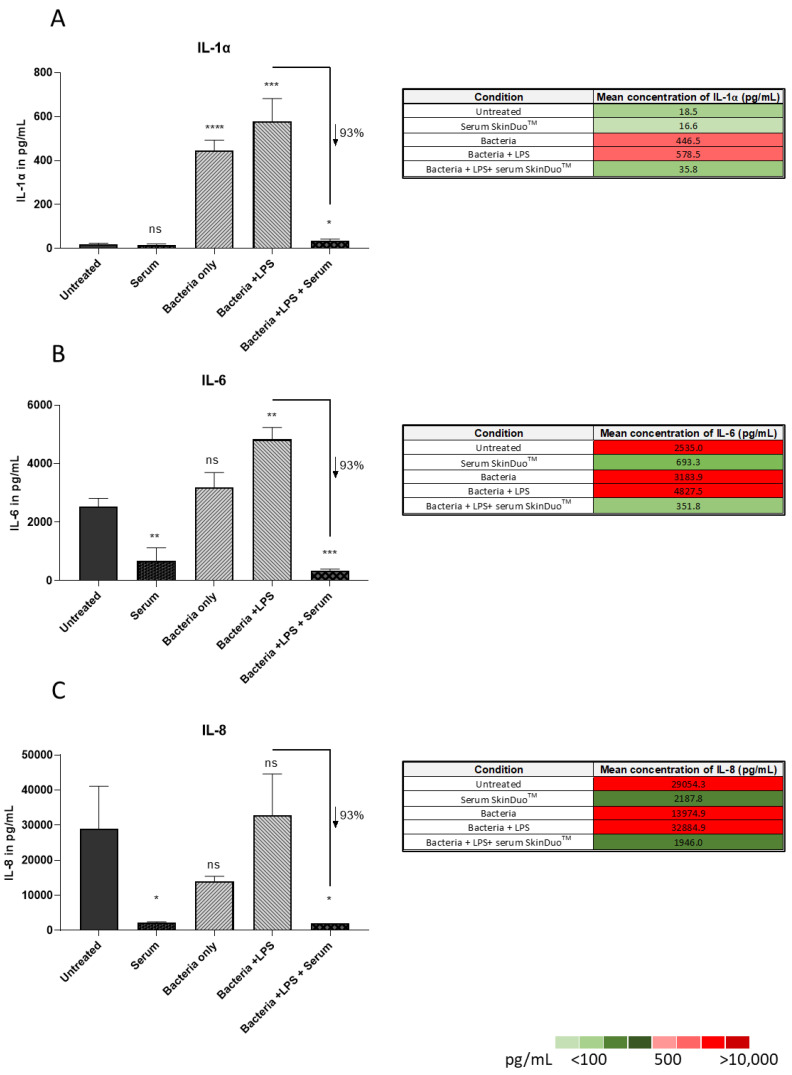
Interleukin measurements by ELISA significantly decreased after serum *L. plantarum* SkinDuo^TM^ treatment. After 48 h of serum (SkinDuo^TM^) application to the co-culture (sebocytes/bacteria: *C. acnes* and *S. epidermidis*), culture supernatants were collected, and interleukin assays were performed. The conditions tested were for: untreated, serum SkinDuo^TM^, bacteria, bacteria + LPS, bacteria + LPS + serum (**A**–**C**). The serum SkinDuo^TM^ statistically reduced the levels of IL-1 α, IL-6 and IL-8 when compared to the in vitro human cellular model for acne and challenged with LPS. *t*-test untreated (sebocytes) Vs.: serum (SkinDuo^TM^), bacteria only, bacteria + LPS, bacteria + LPS + serum. Unpaired *t*-test: 0.1234 (NS), 0.0332 (*), 0.021 (**), 0.0002 (***), <0.0001 (****), percentage decrease of serum SkinDuo^TM^ treatment compared to bacteria challenged with LPS. Table colour intensity (adjacent to each graph) represents a scale intensity from green to red (lowest to highest concentration).

## Data Availability

Raw data available upon request.

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
