# Peer review of "Topical Administration of Lactiplantibacillus plantarum (SkinDuoTM) Serum Improves Anti-Acne Properties"

_microorganisms, 2023, doi:10.3390/microorganisms11020417_

Round 1

Reviewer 1 Report

The experiments are well conducted and my recommendations are the following:

-          Line 94: Sebocytes, I suggestsebocytes” (I marked with yellow)

-          Line 104: 2mL”, I suggest a blank, “2 mL” (I marked with yellow)

-          Lines 106, 107, 113, 134: “37°C”, I suggest a blank, “37 °C” (I marked with yellow)

-          Line 108: “48h”, I suggest a blank, “48 h” (I marked with yellow)

-          Line 113: “1mL”, I suggest a blank, “1 mL” (I marked with yellow)

-          Line 134: “72h”, I suggest a blank, “72 h” (I marked with yellow)

-          Line 142: “50μl”, “7μl”, I suggest, “50 μL”, “7 μL” (I marked with yellow)

-          Line 163: “570nm”, “585nm”, I suggest a blank, “570 nm”, “585 nm” (I marked with yellow)

-          Lines 168: “-80°C”, I suggest a blank, “-80 °C” (I marked with yellow)

-          Line 176: “800μl”, I suggest a blank, “800 μL” (I marked with yellow)

-          Line 177: “400 μl”, I suggest, “400 μL” (I marked with yellow)

-          Line 177: “2,500rpm”, I suggest a blank, “2,500 rpm” (I marked with yellow)

-          Line 178: “500μl”, I suggest a blank, “500 μL” (I marked with yellow)

-          Line 186: “70°C”, „5min”, I suggest a blank, “70 °C” “5 min”(I marked with yellow)

-          Line 187: “25μM”, I suggest a blank, “25 μM” (I marked with yellow)

-          Lines 200, 201: “2cm”, I suggest a blank, “2 cm” (I marked with yellow)

-          Line 201: “1.5mls”, I suggest, “1.5 mL”,  (I marked with yellow)

-          Lines 201, 243: “4°C”, I suggest, “4 °C” (I marked with yellow)

-          Lines 270, 271: “8h, 24h and 48h”, I suggest blank, “8 h, 24 h and 48 h” (I marked with yellow)

-          Line 272: “8mm”, I suggest, “8 mm” (I marked with yellow)

-          Line 276: “48h”, I suggest, “48 h” (I marked with yellow)

-          Line 278: “2cm2 x 2cm2”, I suggest blank, “2 cm2 x 2 cm2” (I marked with yellow)

-          Line 320: “32884.9pg/mL”, I suggest blank, “32884.9 pg/mL” (I marked with yellow)

-          Line 321: “35.8pg/mL, 351.8pg/mL 1946pg/mL”, I suggest blank, “35.8 pg/mL, 351.8 pg/mL 1946 pg/mL” (I marked with yellow)

-          Line 331: “8h, 48h, 72h, 96h, 120h”, I suggest blank, “8 h, 48 h, 72 h, 96 h, 120 h” (I marked with yellow)

-          Line 332: “24h”, I suggest blank, “24 h” (I marked with yellow)

-          Line 334: “24, 48, 72h”, I suggest blank, “24 h, 48 h, 72 h” (I marked with yellow)

-          Line 340: “8h”, “24h”, “48h”, I suggest blank, “8 h”, “24 h”, “48 h” (I marked with yellow)

-          Line 344: “T8”, I suggest, “T8h” (I marked with yellow)

-          Line 350: “8h”, I suggest blank, “8 h” (I marked with yellow)

-          Line 354: “48hours”, I suggest blank, “48 hours” (I marked with yellow)

-     At IntroductionI suggest the use of the neuter form (“were studied....”) and not the first plural form (“we studied.....”) (I marked with yellow)

-     At 5. Conclusions, I suggest the use of the neuter form (“The obtained results….”) and not the first plural form (“Our results…”) (I marked with yellow)

-    In all manuscript it is necessary to use Italic Font for the latin names of microorganisms species. (I marked with yellow)

-  Please, I recommend to verify all citations from References in accordance with MDPI Journals requirements.

-    References should be described as follows, depending on the type of work: https://www.mdpi.com/authors/references    

Author Response

Dear Reviewer 1,

We thank the reviewer 1 for the useful comments and we are pleased to have commented and edited each point that has been raised. The figures have been edited to conform with the layout of the manuscript, discussion and conclusion have been edited, as well as legends from each figure.

  •          Line 94: „Sebocytes”, I suggest “sebocytes” (I marked with yellow). This has been corrected.

-          Line 104: „2mL”, I suggest a blank, “2 mL” (I marked with yellow). This has been corrected.

-          Lines 106, 107, 113, 134: “37°C”, I suggest a blank, “37 °C” (I marked with yellow).This has been corrected.

-          Line 108: “48h”, I suggest a blank, “48 h” (I marked with yellow). This has been corrected.

-          Line 113: “1mL”, I suggest a blank, “1 mL” (I marked with yellow). This has been corrected.

-          Line 134: “72h”, I suggest a blank, “72 h” (I marked with yellow). This has been corrected.

-          Line 142: “50μl”, “7μl”, I suggest, “50 μL”, “7 μL” (I marked with yellow). This has been corrected.

-          Line 163: “570nm”, “585nm”, I suggest a blank, “570 nm”, “585 nm” (I marked with yellow). This has been corrected.

-          Lines 168: “-80°C”, I suggest a blank, “-80 °C” (I marked with yellow). This has been corrected.

-          Line 176: “800μl”, I suggest a blank, “800 μL” (I marked with yellow). This has been corrected.

-          Line 177: “400 μl”, I suggest, “400 μL” (I marked with yellow). This has been corrected.

-          Line 177: “2,500rpm”, I suggest a blank, “2,500 rpm” (I marked with yellow). This has been corrected.

-          Line 178: “500μl”, I suggest a blank, “500 μL” (I marked with yellow). This has been corrected.

-          Line 186: “70°C”, „5min”, I suggest a blank, “70 °C” “5 min”(I marked with yellow). This has been corrected.

-          Line 187: “25μM”, I suggest a blank, “25 μM” (I marked with yellow). This has been corrected.

-          Lines 200, 201: “2cm”, I suggest a blank, “2 cm” (I marked with yellow). This has been corrected.

-          Line 201: “1.5mls”, I suggest, “1.5 mL”,  (I marked with yellow). This has been corrected.

-          Lines 201, 243: “4°C”, I suggest, “4 °C” (I marked with yellow). This has been corrected.

-          Lines 270, 271: “8h, 24h and 48h”, I suggest blank, “8 h, 24 h and 48 h” (I marked with yellow). This has been corrected.

-          Line 272: “8mm”, I suggest, “8 mm” (I marked with yellow). This has been corrected.

-          Line 276: “48h”, I suggest, “48 h” (I marked with yellow). This has been corrected.

-          Line 278: “2cm2 x 2cm2”, I suggest blank, “2 cm2 x 2 cm2” (I marked with yellow). This has been corrected.

-          Line 320: “32884.9pg/mL”, I suggest blank, “32884.9 pg/mL” (I marked with yellow). This has been corrected.

-          Line 321: “35.8pg/mL, 351.8pg/mL 1946pg/mL”, I suggest blank, “35.8 pg/mL, 351.8 pg/mL 1946 pg/mL” (I marked with yellow). This has been corrected.

-          Line 331: “8h, 48h, 72h, 96h, 120h”, I suggest blank, “8 h, 48 h, 72 h, 96 h, 120 h” (I marked with yellow). This has been corrected.

-          Line 332: “24h”, I suggest blank, “24 h” (I marked with yellow). This has been corrected.

-          Line 334: “24, 48, 72h”, I suggest blank, “24 h, 48 h, 72 h” (I marked with yellow). This has been corrected.

-          Line 340: “8h”, “24h”, “48h”, I suggest blank, “8 h”, “24 h”, “48 h” (I marked with yellow). This has been corrected.

-          Line 344: “T8”, I suggest, “T8h” (I marked with yellow). This has been corrected.

-          Line 350: “8h”, I suggest blank, “8 h” (I marked with yellow). This has been corrected.

-          Line 354: “48hours”, I suggest blank, “48 hours” (I marked with yellow). This has been corrected.

-     At IntroductionI suggest the use of the neuter form (“were studied....”) and not the first plural form (“we studied.....”) (I marked with yellow). This has been corrected with neuter form.

-     At 5. ConclusionsI suggest the use of the neuter form (“The obtained results….”) and not the first plural form (“Our results…”) (I marked with yellow). This has been corrected.

-    In all manuscript it is necessary to use Italic Font for the latin names of microorganisms species. (I marked with yellow). This has been corrected throughout the manuscript inclusive title of section 3.1.1.

  • Please, I recommend to verify all citations from References in accordance with MDPI Journals requirements. The references have been all corrected and updated as required by MDPI Journals.  The references are now updated in accordance with MDPI Journal requirements. 

Reviewer 2 Report

Microorganisms (ISSN 2076-2607)

 Christine Podrini et al. has investigated topical administration of Lactiplantibacillus plantarum (SkinDuoTM) serum improves anti-acne properties. Has been developed an innovative anti-acne serum with L. plantarum that mimics over production of lipids, anti-inflammatory properties and improves acne-disease skin models. Based on these results, the authors suggest that SkinDuoTM may be introduced as an acne-mitigating agent. The authors present an original and interesting work about Lactiplantibacillus plantarum (SkinDuoTM) serum. The study design is well rationalized and thorough, and supported by adequate data. In general, the manuscript of Christine Podrini et al. contains relevant paragraphs that have been discussed. The selection of bibliography is appropriate to the content of the manuscript. In the discussion the authors have included short conclusion. Perhaps even too brief conclusion. Finally, regarding methodology, authors  refer about statistics thus the readers can make assumptions regarding the quality and the confidence of the results and reasonability of consideration of the authors.

 Dear authors: Topical administration of Lactiplantibacillus plantarum (SkinDuoTM) serum improves anti-acne properties. In general, the study is closely connected to the journal's objectives. The study is very interesting. The abstract is good. The introduction has updated references until 2019. Overall, the work is interesting and will be beneficial for the fellow researchers. I will recommend it after some major revisions.

 The section and subsections need to improve (3. Results

3.1

3.1.1. Rationale for Lactiplantibacillus plantarum).

References section needs major improvement:

for example:

28. Lomholt, H. B., Scholz, C. F. P., Bruggemann, H., Tettelin, H. & Kilian, M. A comparative study of Cutibacterium 537 (Propionibacterium) acnes clones from acne patients and healthy controls. Anaerobe 47, 57–63 (2017).

39. Jugeau S, Tenaud I, Knol AC, Jarrousse V, Quereux G, Khammari A, et al. Induction of toll-like receptors by 564 Propionibacterium acnes. Br J Dermatol. 2005; 153(6): 1105 – 1113.

 After an exhaustive revision, the manuscript is Reconsider after minor revision.

Author Response

Dear Reviewer 2,

We appreciate your useful comments for the manuscript submitted. We have responded to your comments and edited the following sections as requested. 

Rationale for Lactiplantibacillus plantarum).

This section has been edited and improved. We hope that this sections is now clearer to the reviewer. Figures are also re edited to be more consistent for all the manuscript.

References section needs major improvement:

The references have been corrected and updated as required by the journal.

We are grateful to the comments of reviewer 2. We have edited the text in section 3.1.1, discussion and conclusions. The figures have been replaced by a new version of graphpad version 9 and legends have been updated with statistical significance. We hope that the edited manuscript is now satisfactory. 

The Discussion has been elaborated and discussed results which were not previously included. 

Reviewer 3 Report

Article interesting and overall well written.

BUT, before being accepted for publications some aspects must be improved (both in text and in figures):

- TEXT:

-Please review sentence at line 286 as you are not studying the whole microbiome, but only single strains. Nothing can be said about the behaviour of the microbiome

-line 385: antibiotics are pehraps not effective on the long run, but I doubt that any doctor would give them if they would be totally ineffective as this sentence seems to imply

- discuussion should be improved: it is based on 3 paragraphs: 2 of introduction and 1 that seems more a conclusion of the study rather than the discussion of the results obtained in the study

- line 246: as previously described: please add reference

- sometimes strains are not in italic, please review text (ex title of 3.1.1, line 252, fig. 2a,...)

-small typos: line 65: ; should be .    

251: tenfold should be ten-fold

line 260: al., remove comme

For the figures:

-please, as described in guide to authors, put figures immediately after their citations and not in a separate section

-FIG1: please reduce overall size , but not of picture,

Fig1:T8,T24, T48 and T72 have a pattern when in black, but not the other time points, please give similar look to all time points; moreover totle have different dimension: see for example 1D vs 1A and 1B, size of title should be similar

-Fig2: legend to images should be reviewed as the colors of neutral lipids, total lipids and nuclei do not seem to correspond to anything in the images/ size bar should be within images

-Fig3: why are there different colors in the tables? Please provide a scale bar that relats data with green or red intensity, and add comment in legend

- figures in general can also be split in smaller ones

Author Response

Dear Reviewer 3,

We thank the reviewer 3 for the useful comments and we have edited the sections required as suggested. Please find below the answers to the raised points.

Reviewer 3:

Line 286 as you are not studying the whole microbiome, but only single strains. Nothing can be said about the behaviour of the microbiome.

This has changed now line 326 to: To the inhibitory activity of lactobacilli against acne-causing bacteria,

-line 385: antibiotics are pehraps not effective on the long run, but I doubt that any doctor would give them if they would be totally ineffective as this sentence seems to imply

This has changed now line 358: even though prolonged treatment may result in not an effective cure.

discuussion should be improved: it is based on 3 paragraphs: 2 of introduction and 1 that seems more a conclusion of the study rather than the discussion of the results obtained in the study.

The discussion has been improved and extra paragraph has been added describing the results obtained from line 492-501 and 516-219. Also the Conclusions, this was edited and improved line 528-541.

line 246: as previously described: please add reference

This sentence was confusing as we wanted to refer to the previous figure. This has been edited now line 250-253, we hope now the revised sentence is clearer.

sometimes strains are not in italic, please review text (ex title of 3.1.1, line 252, fig. 2a,...)

All strains in the text are now in italics inclusive title 3.1.1 and line 252, and Figure 2a. 

-small typos: line 65: ; should be .    corrected

251: tenfold should be ten-fold  corrected

line 260: al., remove comme corrected

For the figures:

-please, as described in guide to authors, put figures immediately after their citations and not in a separate section

corrected

-FIG1: please reduce overall size , but not of picture,

corrected

Fig1:T8,T24, T48 and T72 have a pattern when in black, but not the other time points, please give similar look to all time points; moreover totle have different dimension: see for example 1D vs 1A and 1B, size of title should be similar

corrected, the figures have all similar looks and fonts. Figure 1 is now split in two figures and no patterns was used as time-points are represented. Figure 3 and Figure 4 patterns are included because they represent different conditions.

-Fig2: legend to images should be reviewed as the colors of neutral lipids, total lipids and nuclei do not seem to correspond to anything in the images/ size bar should be within images

Now Figure 3, apologies the represented images are the merged images for Nile red neutral lipid FITC, total lipid TRITC and nuclei DAPI, presented as yellow lipid droplets. The graph presents the ratio between the total lipid/nuclei for each condition in triplicates, the scale bar is included in each image. The serum probiotic  diminishes the ratio of total lipids/nuclei when compared to induced bacteria with LPS and as presented in the images.

-Fig3: why are there different colors in the tables? Please provide a scale bar that relats data with green or red intensity, and add comment in legend

Comments added to the legend, colour change corresponds to lowest to highest concentration, scale bar is provided in now Figure 4.

-figures in general can also be split in smaller ones

Figures are now splits to 4 Figures in total, graphs are all edited with Prism Software version 9 and fonts were kept the same throughout the figures. 

We hope that the above corrections are satisfactory for reviewer 3. We are grateful for the useful comments.